# PVDF/Graphene Composite Nanoporous Membranes for Vanadium Flow Batteries

**DOI:** 10.3390/membranes9070089

**Published:** 2019-07-19

**Authors:** Yiming Lai, Lei Wan, Baoguo Wang

**Affiliations:** The state key laboratory of Chemical Engineering, Department of Chemical Engineering, Tsinghua University, Beijing 100084, China

**Keywords:** graphene, nanoporous membrane, ion selectivity, poly(vinylidene fluoride), vanadium flow battery

## Abstract

The development of chemically stable and high conductive membranes is one of the most important issues to improve the performance of vanadium flow batteries (VFBs). Herein, poly(vinylidene fluoride) (PVDF)/graphene composite nanoporous membranes were easily fabricated by manipulating crystallization processes. The graphene was used to enhance membrane selectivity and conductivity. In the nanoscale channels of the membranes, the graphene nanosheets reduced the apertures among the crystal grains, thus restraining vanadium ions crossover due to the size exclusion effect. Moreover, the oxygen groups on the graphene improved the surface hydrophilicity and formed hydrogen bonds with the PVDF polymer chains, which facilitated the proton transport. The composite membranes, with a 0.15 wt % graphene loading, showed a selectivity of 38.2 and conductivity of 37.1 mS/cm. The single cell exhibited a coulomb efficiency of 94.7%, a voltage efficiency of 88.5%, and an energy efficiency of 83.8%, which was 13% higher than that of the pristine PVDF membranes. The composite membranes showed excellent stability during 100 charge-discharge cycles. All these results indicate that the PVDF/graphene composite membrane is a promising candidate for VFB applications.

## 1. Introduction

With the rapid consumption of fossil energy and the profusion of greenhouse gas emissions, more attention has been paid to using renewable energy, such as solar and wind energies [1]. To make full use of intermittent renewable energy sources, large-scale energy storage systems are becoming urgent. Vanadium flow batteries (VFBs), one of the most promising large-scale energy storage devices, have the attractive advantages of intrinsic safety, high efficiency, long service life, as well as environmental friendliness [2,3,4]. A VFB single cell is composed of liquid electrolytes, two electrodes, and an ion conductive membrane (ICM). Two redox couples, VO_2_^+^/VO^2+^ and V^3+^/V^2+^ in the catholyte and anolyte, are separated by ion conductive membranes [5,6].

As a critical component of VFBs, an ICM should prevent multivalent vanadium ions in electrolytes from cross-mixing while transporting protons. The ideal ICMs should have high proton conductivity, low vanadium permeation, good chemical stability, and low cost [7,8]. So far, the most commonly used ICMs in VFBs are ion exchange membranes (IEMs) [9]. As is well-known, the pentavalent vanadium ions is a strong oxidizer, which usually leads to the degradation of ion exchange groups in IEMs [10]. The state-of-the-art ICMs in VFBs have to use Nafion membranes (Dupont) for considering the chemical resistance and proton conductivity, although it also suffers from the limitations of a high cost and a poor selectivity to vanadium ions in VFB applications [11]. Nowadays, numerous ion exchange membranes (IEMs) have been prepared and investigated, including modified Nafion [12,13,14], polybenzimidazole (PBI) [15,16,17], and sulfonated poly(ether ether ketone) (SPEEK) membranes [18,19].

To improve the membrane selectivity for decreasing cross contamination, many inorganic nanomaterials have been exploited to add into polymer matrix of IEMs. These materials include graphene [20], graphene oxide (GO) [4,21], titanium oxide (TiO_2_) [22], and metal organic frameworks (MOFs) [23], playing a critical role as barriers to vanadium ions in the nanoscale channels and effectively reducing the crossover of vanadium ions. Graphene is a typical two-dimensional carbon nanomaterial, with good chemical stability, high surface area, and functional groups [24]. It has received especial attention to prepare IEMs in many researches [20].

In contrast to IEMs, the nanoporous membranes depend on the size exclusion effect to find selectivity, rather than ion exchange mechanism for ion transport [11,25,26]. Their chemical stability obtained significant increase due to their no functional group degradation. In our previous investigation, a controllable crystallization strategy was proposed to prepare PVDF nanoporous membranes [27], where the structure of the membranes could be tuned by the composition of nucleating agent sodium allyl sulfonate (SAS), crystallization time, and processing temperature. This composite membrane has been used to make a 6 kW VFBs stack, with energy efficiency about 80% during 650 charge/discharge cycles. These results demonstrated that the PVDF membrane can provide enough stability. So far, some modification space still remains to improve its conductivity and vanadium ion selectivity to attain a better application.

In this study, we prepared a PVDF/graphene composite membrane using a controllable crystallization method, where the crystals in the polymer solution gradually grow and increase in size to form globular grains. These grains approach each other and form a sieve network in the membrane. Ion permeation and selectivity were found to occur by size exclusion and spatial hindrance in the nanoporous membrane, rather than ion exchange and static repulsion effect, thus increasing stability even in harsh oxidative condition of vanadium electrolyte. Moreover, graphene nanosheets were uniformly dispersed among polymer spherulites to reduce the pore size of the nanoscale channels and improve the surface hydrophilicity by the formation of hydrogen bonds. The membrane properties were characterized by Fourier transform infrared spectroscopy (FT-IR), scanning electron microscopy (SEM), ion selectivity, proton conductivity, and single flow battery test. Results showed that the PVDF/graphene composite membrane could be a promising candidate for VFBs.

## 2. Materials and Methods

### 2.1. Materials

Poly(vinylidene fluoride) (PVDF, FR904) was purchased from 3F New Material Co., Ltd., Shanghai, China. Sodium allyl sulfonate (SAS, 97 wt %) was provided by Ouhe Chemical Technology Co., Ltd., Beijing, China. Dimethyl sulfoxide (DMSO) was supplied by Shanghai Aladdin Bio-Chem Technology Co., Ltd., Shanghai, China. Graphene (G250-H) was purchased from Tanmei Technology Co., Ltd., Taiyuan, China. Sulfuric acid was purchased from Beijing Chemical Works, Beijing, China. Vanadyl sulfate (VOSO_4_) hydrate was purchased from Shenyang Haizhongtian Fine Chemical Factory, Shenyang, China. All reagents were used without further purification.

### 2.2. Membrane Preparation

The composite PVDF nanoporous membranes were prepared through a brief solution casting method. First, a certain amount of graphene was dispersed in DMSO (10 g) under sonication for 5 h. PVDF (15 g) and SAS (5 g) were dissolved in DMSO (70 g) with continuous stirring at 80 °C for 4 h to obtain a homogeneous solution. Then, the graphene suspension was transferred to the PVDF/SAS/DMSO solution and the mixture was vigorously stirred for 24 h at 80 °C to a uniform casting solution. The mixture was kept at 80 °C for 8 h to eliminate the bubbles. Afterward, the mixture was cast onto a clean glass plate and evaporated at 80 °C for 6 h. The composite membrane was soaked in deionized water to remove the SAS nucleating agent. Finally, the fabricated membrane was stored in deionized water before use.

### 2.3. Membrane Characterization

#### 2.3.1. Membrane Morphology

The cross-sectional morphology of membranes was observed by field emission scanning electron microscopy (FESEM) (Merlin, Carl Zeiss, Oberkochen, Germany). To investigate the cross-sections, the samples were fractured in liquid nitrogen. Before observation, all the samples were sputter coated with carbon.

#### 2.3.2. Ion Permeability and Selectivity

To determine the permeability and selectivity of proton and vanadium ions, a diffusion cell, which sandwiched the PVDF nanoporous membranes with two chambers, was designed. The effective diffusion area was 7.065 cm^2^ (a circular cross section, with a diameter of 3 cm). The left chamber was filled with a 1.5 M VOSO_4_ and 3 M H_2_SO_4_ aqueous solution (35 mL), while the right chamber was filled with deionized water (35 mL). During the measurement, both sides were stirred vigorously to avoid concentration polarization. The concentration of H^+^ in the right chamber was detected by a pH meter (S220 Sevencompact, Mettler Toledo, Giessen, Germany), while the VO^2+^ concentration was measured by a UV-vis spectrophotometer (UV-1800PC, MAPADA, Shanghai, China). The concentrations of H^+^ and VO^2+^ were recorded every second and 30 min, respectively.

The ion permeability was determined by the following equation according to Fick’s diffusion law:(1)$$V_{B}\frac{{\mathrm{dC}}_{B}\left(t\right)}{\mathrm{dt}}\;=\;A\frac{P}{L}{(C}_{A}{(t)\;-\;C}_{B}\left(t)\right)$$
where *V*_B_ is the volume of the solution in the right chamber; A is the effective diffusion area; L is the thickness of the membrane; *C*_A_(*t*) and *C*_B_(*t*) are the ion concentrations in the left and right chamber, respectively, as functions of the diffusion time, *t*; P is the permeability of ions in the membrane.

Given that the volume of solution was kept constant during the diffusion process,
(2)$$C_{A}{(t)\;+\;C}_{B}{(t)\;=\;C}_{A}\left(0\right)$$
where *C*_A_(0) is the original ion concentration in the left chamber, then,
(3)$$P\;=\;-\frac{dV_{B}}{2A}\frac{\mathrm{dln}[1/2C_{A}{(0)\;-\;C}_{B}\left(t)\right]}{\mathrm{dt}}$$

The ion selectivity is defined as the ratio of the permeability,
(4)$$H/V\;\mathrm{selectivity}\;=\;\frac{{P(H}^{+})}{{P(\mathrm{VO}}^{2+})}$$

#### 2.3.3. Area Resistance and Conductivity

The area resistance of the membranes was measured using a conductivity cell composed of two chambers, each containing a 2 M H_2_SO_4_ aqueous solution. Two platinum electrodes were set at a fixed constant distance. The resistance of the cell with and without a membrane was measured by electrochemical impedance spectroscopy (EIS) using an electrochemistry workstation (VersaSTAT3, Princeton, NJ, USA), over a frequency range of 1 MHz to 1 Hz at room temperature. The area resistance *r* was calculated by the following equation:(5)$${r\;=\;(r}_{1}{\;-\;r}_{2})\;\times \;S$$
where *r*_1_, *r*_2_ are the cell resistance with and without membrane, respectively, and S is the effective area.

The proton conductivity was calculated as follows:(6)$$\sigma \;=\;\frac{L}{r}$$

#### 2.3.4. Attenuated Total Reflectance Fourier Transform Infrared Spectroscopy (ATR-FTIR) 

The prepared PVDF nanoporous membranes and PVDF/graphene composite membranes were characterized by attenuated total reflectance (ATR) (Nicolet 6700FTIR, Thermo Electron Corporation, Shanghai, China). The chemical composition of graphene was characterized by FT-IR (Nicolet 6700FTIR, Thermo Electron Corporation, Shanghai, China). Each spectrum was obtained by frequency scanning from 400 to 4000 cm^−1^_._

#### 2.3.5. VFB Performance

A VFB single cell was assembled by sandwiching the fabricated membrane between two carbon felts with an effective area of 25 cm^2^. The carbon felts were used as positive and negative electrodes with 33% compression. Two graphite plates were used as current collectors: 1.5 M VO^2+^/VO_2_^+^ and 1.5 M V^2+^/V^3+^ in 3 M H_2_SO_4_ solution were used as the anolyte and catholyte, respectively. The electrolytes were separately cycled by two magnetic pumps, with the protection of N_2_. The charge-discharge tests were performed by a battery test system (CT2001B, Land, Wuhan, China). For cycle tests at certain current densities, the cut-off voltages of the charge and discharge processes were set to 1.65 and 0.8 V, respectively. The coulomb efficiency (CE), voltage efficiency (VE), and energy efficiency (EE) were calculated by the following equations:(7)$$\mathrm{CE}\;=\;\frac{\int I_{d}\mathrm{dt}}{\int I_{c}\mathrm{dt}}\;\times \;100\%$$
(8)$$\mathrm{CE}\;=\;\frac{\int V_{d}\mathrm{dt}}{\int V_{c}\mathrm{dt}}\;\times \;100\%$$
(9)EE = CE × VE
where *t* is time, *V*_d_ and *V*_c_ are voltages of the discharge and charge processes, and *I*_d_ and *I*_c_ are currents of the discharge and charge processes, respectively. 

#### 2.3.6. Mechanical Properties

The mechanical properties were studied by tensile testing. The membranes were shaped as 150 mm× 15 mm, with a clamp distance of 100 mm at a stretching speed of 100 mm/min. 

## 3. Results and Discussion

### 3.1. Characterization of PVDF and PVDF/Graphene Composite Membranes

Pristine PVDF membranes with different contents of SAS were prepared, we characterized the ion selectivity and proton conductivity (Appendix A). The ion selectivity of the PVDF nanoporous membranes decreased with the content of nucleating agent, while the proton conductivity increased. The optimized membrane with 25 wt % SAS exhibited a high ion selectivity of 26.6 and proton conductivity of 23.5 mS/cm. 

Furthermore, PVDF/G-m (m = 0.05, 0.10, 0.15, 0.2, 0.3 wt %) composite membranes with 25 wt % SAS content were prepared and characterized to illustrate the effect of the graphene nanosheets. A digital photograph of the pristine PVDF membrane and the synthesized PVDF/G-0.15 composite membrane is shown in Figure 1. With the addition of graphene, the color of the membranes changed from almost colorless to deep black. The picture also shows that the graphene nanosheets were uniformly dispersed in the PVDF matrix. As shown in Figure 2, the cross-sectional morphology of the PVDF and PVDF/G-0.15 composite membranes were overall dense, homogeneous, and symmetrical. Compared with a pristine PVDF membrane, the wrinkles in the cross-section morphology of PVDF/G-0.15 were smaller and distributed more evenly over the whole section. It was also demonstrated that graphene was dispersed uniformly in the PVDF polymer substrate, without obvious agglomeration, owing to the physical dispersion and the interaction between graphene and the PVDF polymer chains.

The morphologies of PVDF/G-0.15 with different crystallization times are shown in Figure 3. The morphology of the cross section changed from a loose and asymmetric structure to a dense and symmetric structure. In the evaporating solvent process, the PVDF molecular chain first crystallized and formed spherulites in the presence of a nucleating agent. The spherulites of PVDF grew and the space between them gradually decreased. After 1 h, the boundary between spherulites disappeared and the gaps between these spherulites decreased to the sub-micron scale. Finally, these gaps formed the membrane pores. As shown in the high-resolution SEM image of the cross-section morphology with a crystallization time of 30 min (Figure 3b), the graphene nanosheets were dispersed uniformly and located in the space between the spherulites. Hence, the graphene decreased the pore size and influenced the physicochemical and electrochemical properties. 

The chemical composition of pristine PVDF membranes, PVDF/graphene-0.15 composite membranes, and high-activity graphene were investigated by ATR and FT-IR. The absorption bands at 1167 and 1230 cm^−1^ were assigned to F–C–F, which indicated the presence of a PVDF matrix [28]. There were no absorption bands at 1280 cm^−1^ (asymmetric stretching of O=S=O), 1010 cm^−1^ (S=O stretching) and 707 cm^−1^ (S–O stretching). This indicated that no SO_3_^−^ existed in the pristine PVDF and PVDF/G-0.15 membranes, and the nucleating agent was completely removed during immersion [27]. This proved that no ion exchange groups existed and the mechanism of ion selectivity was pore size exclusion. High-activity graphene was introduced to the PVDF polymer matrix in the composite membranes, and the FT-IR spectrum of graphene is shown in Figure 4. The characteristic absorbance peak at 1726 cm^−1^ corresponded to C=O stretching of the carbonyl (C=O) and carboxylic (–COOH) groups. The peak at 1219 cm^−1^ showed the existence of epoxy groups. The wide absorption band in the range of 3300–2500 cm^−1^ corresponded to the O-H vibration in –COOH. The graphene with hydrophilic oxygen-containing groups and the PVDF matrix constituted proton transport channels at the interface. The oxygen functional groups improved the hydrophilicity of the interface. In addition, these oxygen groups on the basal plane and at the edge of the graphene nanosheets formed hydrogen bonds with the fluorine on the PVDF polymer chains and water [29,30,31]. This interaction could improve the proton conduction through the membrane, according to the Grotthuss mechanism, proton hopping through a hydrogen-bonding network [32,33].

As for the mechanical properties, Appendix A shows that the incorporation of graphene had little influence on the tensile strength of the membranes. The PVDF/G-0.15 membranes had nearly the same tensile strength as the PVDF membranes, which ensured the reliability in VFB applications.

For an ICM aimed at a commercial flow battery application, ion selectivity and proton conductivity are two significant parameters. The selectivity indicates an ion selective permeability and a pore size distribution of the membranes. The results are shown in Table 1 and Figure 5. With the graphene loading content increasing from 0% to 0.15%, the VO^2+^ permeability decreased from 12.8 × 10^−7^ to 7.09 × 10^−7^ cm^2^/min, and then was almost kept constant. First, the graphene nanosheets were uniformly dispersed in a casting solution. During solvent evaporation, the PVDF polymer began to crystalize and formed spherulites with the nucleating agents. The spherulites grew and the space where the graphene nanosheets remained, between the spherulites, was decreased to form nanoscale pores. Hence, the graphene made the nanopores smaller and reduced the number of defects. Tetravalent vanadium ions and protons have different Stokes radii and molecular mass. The Stokes radius of protons (<0.24 nm for H_5_O_2_^+^, an in situ form of H_3_O^+^) is smaller than that of the hydrated multivalent vanadium ions (>0.6 nm for [V(SO_4_)^2−^(H_2_O)_5_]^+^, [VO(H_2_O)_5_]^2+^, and [VO_2_(H_2_O)_3_]^+^) [34,35]. Owing to the exclusion effect of graphene, the permeability of vanadium ions—with a larger Stokes radius—was evidently reduced, while the protons could efficiently pass through the nanoscale pores. However, excessive graphene nanosheets hindered the proton transport and led to a reduction of H^+^ permeability, to 2.50 × 10^−5^ cm^2^/min. As shown in Figure 5, the selectivity, defined as the ratio of H^+^ permeability to VO^2+^ permeability, was parabolic and correlated with the graphene loading content. PVDF/G-0.15 (0.15% graphene loading) exhibited the highest ion selectivity of 38.2 (26.6 for PVDF nanoporous membranes). This optimized graphene loading content gave consideration to both ion permeabilities and achieved a better performance in the further VFB test.

Proton conductivity reflects the migration of a proton through the membranes in an electric field. Using membranes with high conductivity can reduce the internal resistance and improve the efficiency of batteries. With an appropriate graphene loading content, the PVDF/G-0.15 composite membranes had a higher proton conductivity of 37.1 mS/cm, compared with pristine PVDF membranes of 23.5 mS/cm. This indicated that the incorporation of highly active graphene improved the proton migration through membranes. After the crystallization, the graphene nanosheets were located in the nanoscale pores of the membranes. The oxygen functional groups on the graphene nanosheets promoted the formation of hydrogen bonds between the PVDF polymer chains and graphene [29]. Moreover, the oxygen groups could form hydrogen bonds with water, realizing the proton hopping transfer [32]. The interaction between proton and hydrogen-bonding network improved the proton conduction through membranes, corresponding with the Grotthuss mechanism for proton hopping through a hydrogen-bonding network [33]. Additionally, the hydrophilic ion groups of the high-activity graphene nanosheets improved the hydrophilicity of the PVDF nanoscale pores, thus enhancing the proton conductivity [36]. Correspondingly, the proton conductivity of the composite membranes was higher than that of pristine PVDF membranes. However, when the graphene loading content was higher than 0.15 wt %, the proton conductivity began to decrease as a result of the blocking effect. Consequently, an appropriate incorporation of graphene nanosheets enhances the ion selectivity and proton conductivity of PVDF nanoporous membranes. Among these membranes, PVDF/G-0.15 had the highest selectivity of 38.2 as well as the highest proton conductivity of 37.1 mS/cm (Figure 5). This indicated that the PVDF/G-0.15 composite membranes would exhibit a better efficiency in VFBs.

### 3.2. VFB Single Cell Performance

Pristine PVDF membranes and PVDF/G-0.15 composite membranes were assembled into VFB single cells to test the electrochemical performance. The charge-discharge curves at 50 mA/cm^2^ are shown in Figure 6. The two curves were obviously different. Owing to the reduced vanadium ions crossover, the cells assembled with the PVDF/G-0.15 membranes exhibited a higher cell capacity than those with the pristine PVDF membranes. This suggested a longer self-discharge time and higher CE as well. Moreover, cells assembled with the PVDF/G-0.15 membranes showed a higher discharge voltage and a lower charge voltage. The charge-discharge curve of the composite membranes had an obvious intersection point, which did not appear in the curve for the pristine PVDF membranes. These results confirmed that the graphene incorporation enhanced the proton conduction, hence reducing the ohmic polarization and overpotential of batteries. Accordingly, the VE of cells assembled with PVDF/G-0.15 membranes were 88.5%, which was much higher than the pristine membranes of 75.7% at 50 mA/cm^2^ (Figure 7). 

To evaluate the practical vanadium ion crossover through membranes in VFBs, the open circuit voltage (OCV) of the cell was recorded at the beginning of a 50% state of charge. The electrolyte was cycled by pumps during the test. The test was finished when the OCV was less than 0.8 V. As shown in Figure 8, the self-discharge time of the PVDF/G-0.15 composite membranes were approximately 34 h, which is 12 h longer than that of the pristine PVDF membranes. The result illustrated that incorporation of 0.15 wt % graphene effectively reduced the vanadium ions crossover, which was in good agreement with the result obtained from the ion selectivity experiments.

Furthermore, the performance of cells assembled with PVDF and PVDF/G-0.15 membranes under different current densities was tested and compared. For PVDF/G-0.15, as shown in Figure 9, as the current density increased from 50 to 120 mA/cm^2^, the CE increased from 95.4% to 97.7%, while the VE decreased from 88.3% to 74.7%, owing to the shorter charging time and higher polarization, respectively [37]. Therefore, the EE reduced from 84.2% to 73.0%. Cells assembled with the pristine PVDF membranes showed a lower efficiency and could not complete the charge-discharge cycle owing to a high membrane resistance when the current density was above 100 mA/cm^2^. At 80 mA/cm^2^, PVDF/G-0.15 showed a 79.6% EE, while a 65.5% EE was obtained for the PVDF membranes. This significant improvement originated from the high ion selectivity and proton conductivity of the composite membranes.

Considering the strongly acidic and oxidizing conditions in electrolytes, the stability of membranes and steady operation are essential for commercial VFB application. The battery cycle performance of PVDF/G-0.15 was investigated at 50 mA/cm^2^. As shown in Figure 10, the VFB single cell assembled with PVDF/G-0.15 exhibited a stable performance during 100 continuous cycles. Moreover, the cell showed a high CE of 96% and a high EE of 84%, which were much higher than those obtained for the PVDF membranes. No obvious efficiency decay was observed, which indicated the excellent stability of the composite membranes in VFBs.

## 4. Conclusions

Highly active graphene nanosheets were incorporated in nanoporous membranes for VFB application. The physicochemical and electrochemical properties of the composite membranes were investigated. The addition of graphene enhanced the proton conductivity and ion selectivity of nanoporous PVDF membranes. During crystallization in the fabrication process, graphene was located in the nanoscale channels among PVDF spherulites, which reduced the pore size and limited the vanadium ions crossover. Moreover, oxygen groups on the edge of the graphene nanosheets improved the surface hydrophilicity of the channels and formed hydrogen bonds with the PVDF polymer chains, which benefited proton conduction through the membranes. Correspondingly, the battery assembled with PVDF/G-0.15 exhibited an EE of 83.8%, which was 13% higher than that of the pristine PVDF membranes. The composite membranes showed excellent stability in 100 charge-discharge cycles and low cost. In brief, the incorporation of graphene nanosheets into PVDF nanoporous membranes is a facile and effective modification method and is a promising candidate for VFB application.

## Figures and Tables

**Figure 1 membranes-09-00089-f001:**
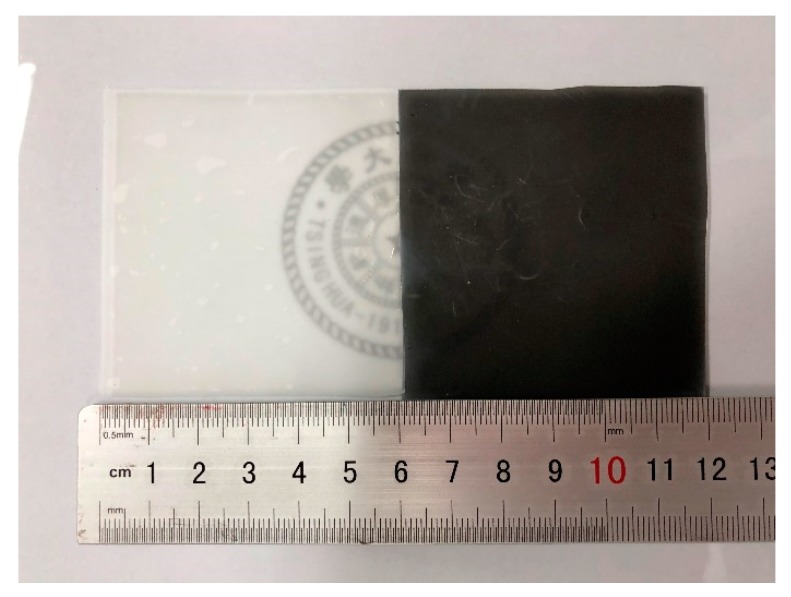
Digital photo of the PVDF nanoporous membrane and the PVDF/G-0.15 membrane.

**Figure 2 membranes-09-00089-f002:**
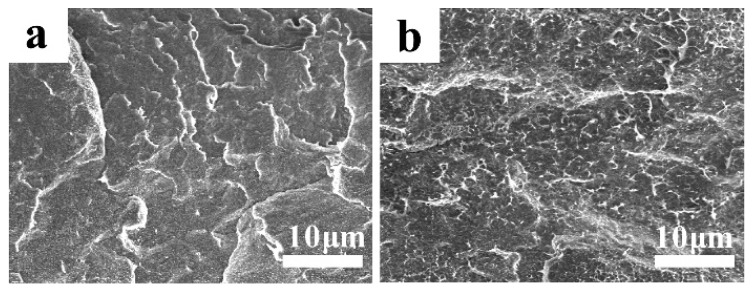
Cross-section morphology of a pristine PVDF membrane (**a**) and composite PVDF membrane with 0.15 wt % of graphene (**b**).

**Figure 3 membranes-09-00089-f003:**
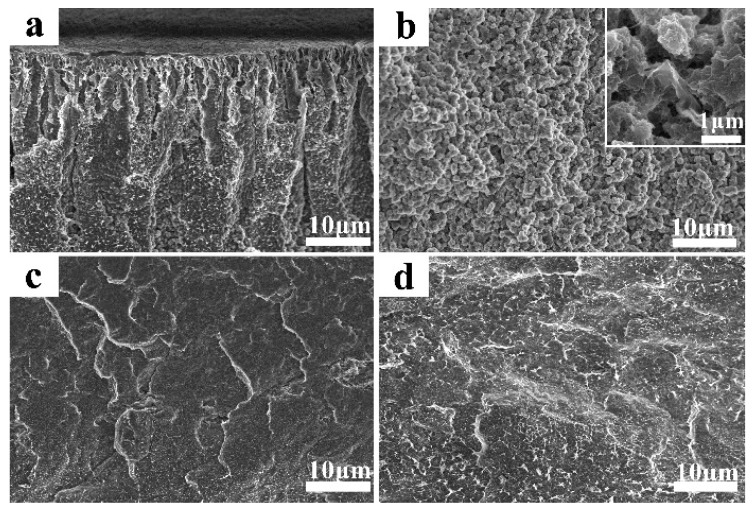
Morphology of PVDF/G-0.15 composite membranes with different crystallization times: (**a**) 10 min; (**b**) 30 min; (**c**) 1 h; (**d**) 6 h.

**Figure 4 membranes-09-00089-f004:**
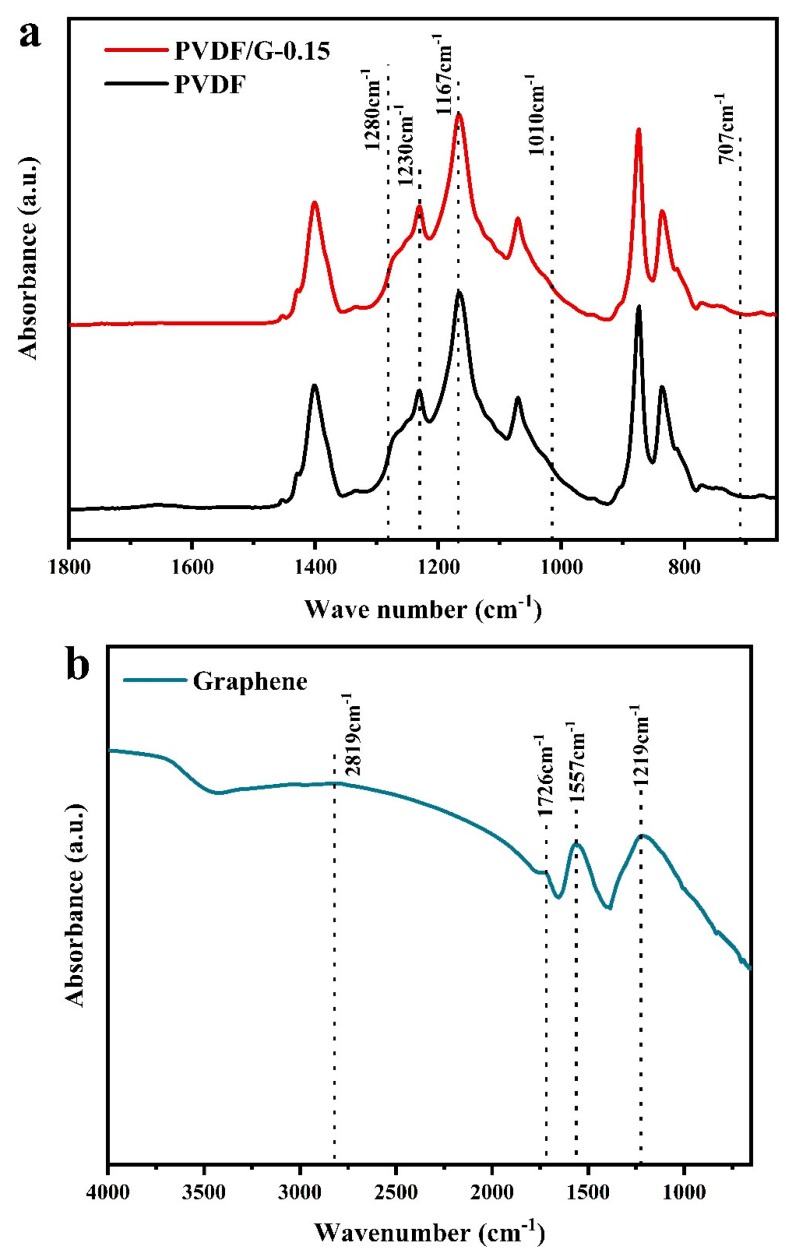
(**a**) ATR spectra of PVDF and PVDF/G-0.15 membranes; (**b**) FT-IR spectrum of the graphene.

**Figure 5 membranes-09-00089-f005:**
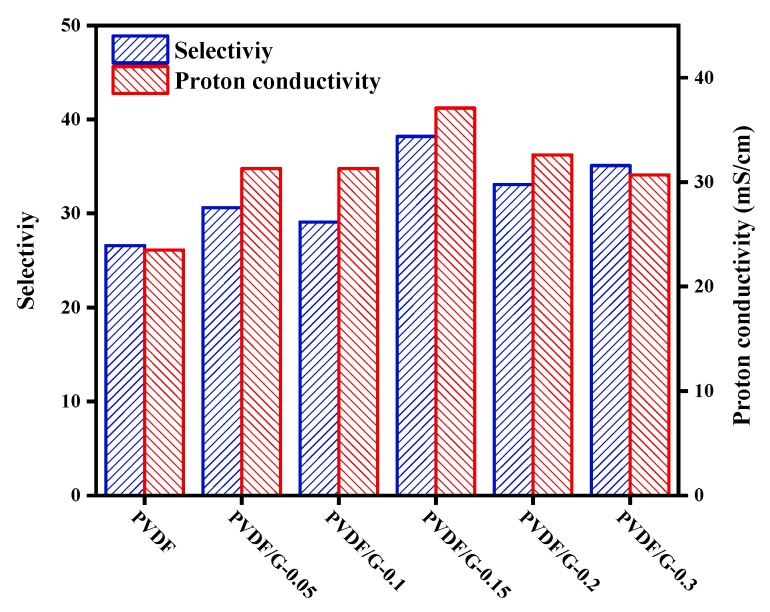
Ion selectivity and proton conductivity of PVDF and PVDF/G membranes.

**Figure 6 membranes-09-00089-f006:**
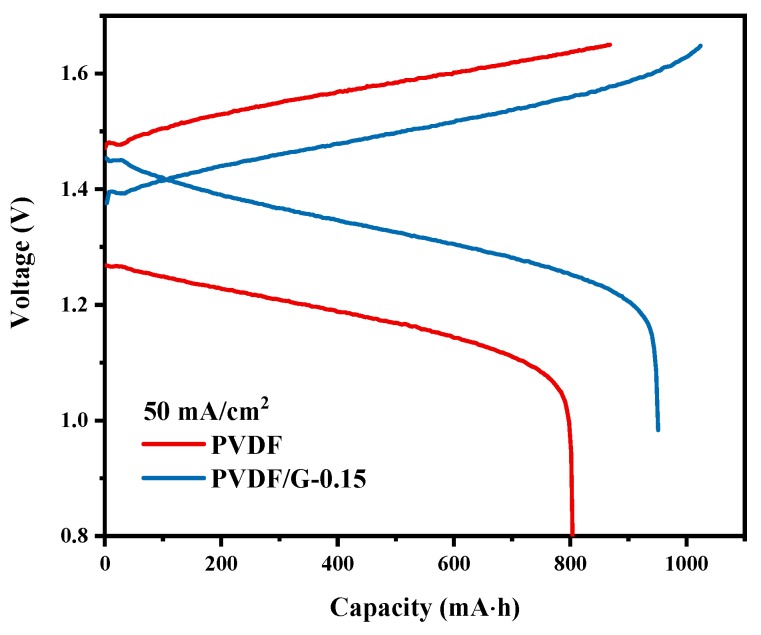
Charge-discharge curves of the VFB single battery assembled with PVDF and PVDF/G-0.15 membranes.

**Figure 7 membranes-09-00089-f007:**
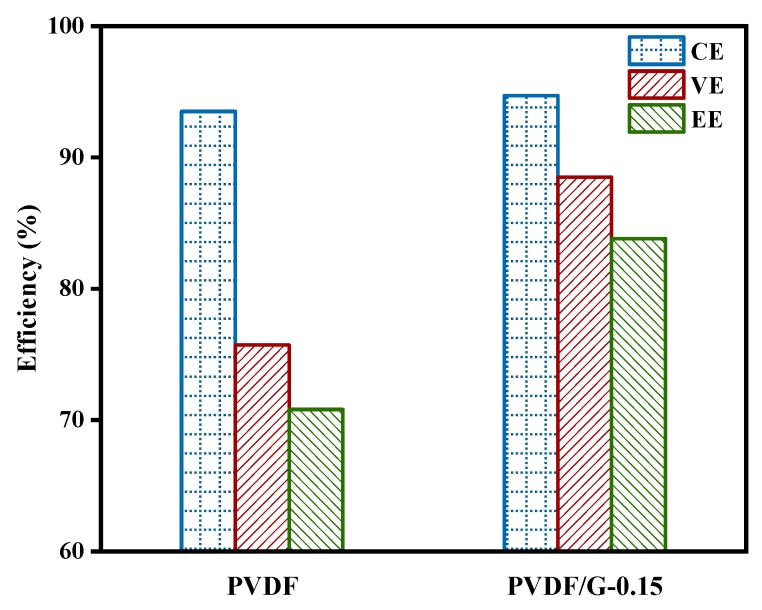
Battery efficiencies assembled with PVDF membranes and PVDF/G-0.15 membranes at 50 mA/cm^2^.

**Figure 8 membranes-09-00089-f008:**
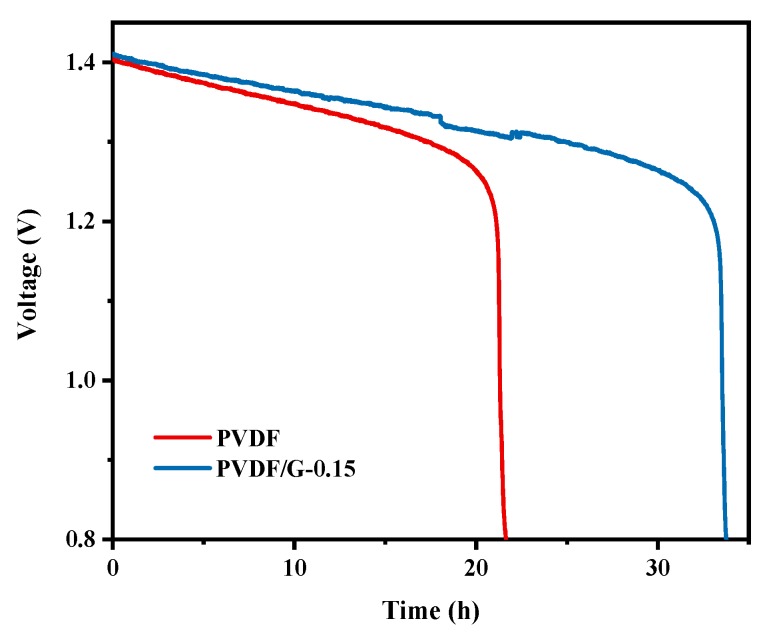
Open circuit voltage of batteries assembled with PVDF and PVDF/G-0.15 membranes.

**Figure 9 membranes-09-00089-f009:**
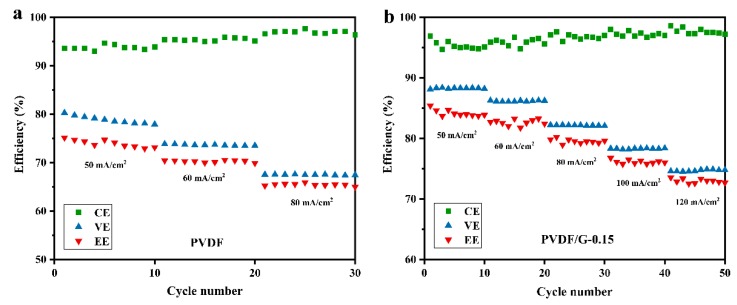
Efficiency variation with current density for different membranes: (**a**) PVDF; (**b**) PVDF/G-0.15.

**Figure 10 membranes-09-00089-f010:**
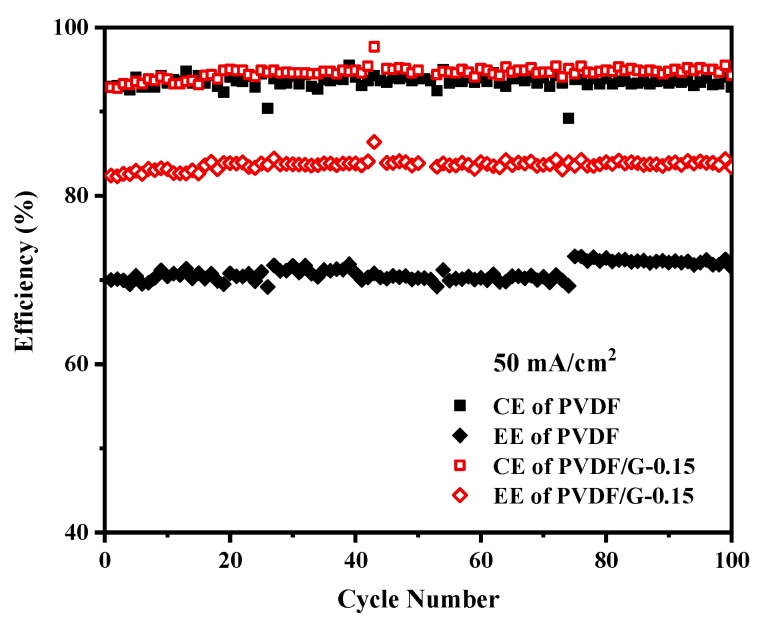
Battery performance assembled with PVDF and PVDF/G-0.15 membranes at 50 mA/cm^2^.

**Table 1 membranes-09-00089-t001:** Physicochemical and electrochemical properties of PVDF and PVDF/G membranes.

Sample	Thickness (μm)	Permeability of Protons (×10^−5^ cm^2^/min)	Permeability of VO^2+^ (×10^−7^ cm^2^/min)	Selectivity	Proton Conductivity (mS/cm)
PVDF	123	3.41	12.8	26.6	23.5
PVDF/G-0.05	124	2.91	9.52	30.6	31.3
PVDF/G-0.10	141	2.85	9.81	29.1	31.3
PVDF/G-0.15	125	3.40	8.89	38.2	37.1
PVDF/G-0.2	115	2.35	7.09	33.1	32.6
PVDF/G-0.3	128	2.50	7.13	35.1	30.7

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
