# Peer review of "PVDF/Graphene Composite Nanoporous Membranes for Vanadium Flow Batteries"

_membranes, 2019, doi:10.3390/membranes9070089_

Round 1

Reviewer 1 Report

The manuscript PVDF/graphene composite nanoporous membranes for vanadium flow batteries by Lai et al. provides very interesting results on proton conductivity and ion selectivity enhancement by graphene incorporation in nanoporous PVDF membrane stractures.

In my opinion the manuscript could be published after the following modifications:

1)      Description of the Figure 2 should be changed: instead of “Cross-section morphology of composite membranes with different graphene contents: (a) 0 wt% and (b) 0.15 wt%.”

The authors should write “Cross-section morphology of a pristine PVDF membrane (a) and composite PVDF membrane with 0.15 wt% of graphene (b).”

2)      Detailed information concerning the FTIR instrument (in the section 2.3.4 Attenuated total reflectance Fourier transform infrared spectroscopy (ATR-FTIR)) should be given.

3)      Description of the Figure 4 should be improved. Instead of “Figure 4. (a) ATR spectra of PVDF membranes and PVDF/G-0.15 membranes; (b) FT-IR spectrum of 213 graphene.” The author should write: “Figure 4. (a) ATR spectra of PVDF and PVDF/G-0.15 membranes; (b) FT-IR spectrum of 213 graphene.” From the current description it looks like they recorded several ATR spectra of each membrane.

4)      It is difficult to read numbers from Figure 4. The size of the figure should be increased, and the Figure 4b should be provide below the Figure 4a, instead of its side.

Author Response

Response to Reviewer 1 Comments

Point 1: Description of the Figure 2 should be changed: instead of “Cross-section morphology of composite membranes with different graphene contents: (a) 0 wt% and (b) 0.15 wt%.”

The authors should write “Cross-section morphology of a pristine PVDF membrane (a) and composite PVDF membrane with 0.15 wt% of graphene (b).”

Response 1: Thank you very much for your suggestions. We have changed the description of Figure 2 from “Cross-section morphology of composite membranes with different graphene contents: (a) 0 wt% and (b) 0.15 wt%.” to “Cross-section morphology of a pristine PVDF membrane (a) and composite PVDF membrane with 0.15 wt% of graphene (b).” in the manuscript.

Point 2: Detailed information concerning the FTIR instrument (in the section 2.3.4 Attenuated total reflectance Fourier transform infrared spectroscopy (ATR-FTIR)) should be given.

Response 2: We have added the detailed information concerning the ATR-FTIR instrument in the manuscript, “Nicolet 6700FTIR, Thermo Electron Corporation, China”.

Point 3: Description of the Figure 4 should be improved. Instead of “Figure 4. (a) ATR spectra of PVDF membranes and PVDF/G-0.15 membranes; (b) FT-IR spectrum of graphene.” The author should write: “Figure 4. (a) ATR spectra of PVDF and PVDF/G-0.15 membranes; (b) FT-IR spectrum of the graphene.” From the current description it looks like they recorded several ATR spectra of each membrane.

Response 3: We have revised the description of Figure 4 in the manuscript, using “Figure 4. (a) ATR spectra of PVDF and PVDF/G-0.15 membranes; (b) FT-IR spectrum of the graphene.”

Point 4: It is difficult to read numbers from Figure 4. The size of the figure should be increased, and the Figure 4b should be provide below the Figure 4a, instead of its side.

Response 4: We have increased the size of these figures and put Figure 4b below Figure 4a in the revised manuscript, as follows.

Reviewer 2 Report

This manuscript presents the synthesis and characterization of novel graphene-PVDF membranes for vanadium redox flow battery applications. Characterization with respect to the properties relevant for VRB applications is carried out very thoroughly. The authors, however, fail to explain the choice of this particular material combination.

I am wondering if it makes sense at all to include an electronic conductor in a material which is supposed to behave as an electronic insulator. Did the authors measure the electronic ASR ? What is the role of the graphene ? Does the ionomer wash out from the membrane ?

Author Response

Thank you very much for your frank comments.

We have revised the manuscript carefully, and give a response point-to-point for your comments. Please see the attached file.

Reviewer 3 Report

The authors report on a novel and promising development of membranes (based on polyvinylidene fluoride / graphene) in vanadium flow batteries. This new conception significantly improves the membranes' ability to minimize exchange of vanadium ions across the conductive membrane  (between anolyte and catolyte). The report is thus an interesting further development of the optimization of vanadium flow batteries.

I recommend publication. The following minor comments might be considered:

(1) Line 81: "vanadyl sulphate hydrate": Commercially available VOSO4 can contain up to 5 molecules of water per formula unit. Has this been taken into consideration when preparing the 1.5 M VOSO4 soln.? This is of some importance in terms of the indication of concentrations such as 1.5 M VOSO4 (line 103). - Also, more correct, following IUPAC nomenclatute, "vanadyl sulfate" is termed "oxidovanadium(IV)sulphate".

(2) Equations (1), (3) and (8): The V = volume should be written in italics (in order to fascilitate distinction from V = vanadium).

(3) Line 142: The vanadium oxidation state +II and +III have been achieved electrochemically (reducton of VO2+)?

4. Line 226: "molecular weight" should be replaced by "molecular mass" (weight is a force).

Author Response

(The authors gave the same response as above.)

Round 2

Reviewer 2 Report

Ms has been revised and a couple of measurements were added.

1) I still would have liked to see the ASR of the dry (!) membranes. 

2) The authors also claim that the graphene improves proton conductivity. How would hat mechanism look like ?

Author Response

Dear Professor

We very appreciate your spending time to evaluate the next manuscript and give frank comments on our study.

Journal: Membranes

Manuscript ID: membranes-544422

Title: PVDF/graphene composite nanoporous membranes for vanadium flow

batteries

Authors: Yiming Lai, Lei Wan, Baoguo Wang*

  Following your comments, we have revised this manuscript seriously and given a response point-by-point, please make a contact with us if you have any questions further.

  Thank you in advance.
